# In Vitro Techniques Using the Daisy^II^ Incubator for the Assessment of Digestibility: A Review

**DOI:** 10.3390/ani10050775

**Published:** 2020-04-29

**Authors:** Sonia Tassone, Riccardo Fortina, Pier Giorgio Peiretti

**Affiliations:** 1Department of Agriculture, Forestry, and Food Sciences, University of Turin, 10095 Grugliasco, Italy; riccardo.fortina@unito.it; 2Institute of Sciences of Food Production, National Research Council, 10095 Grugliasco, Italy; piergiorgio.peiretti@ispa.cnr.it

**Keywords:** in vitro digestibility, inoculum, rumen fluid, faeces, enzyme, Ankom Daisy^II^ incubator

## Abstract

**Simple Summary:**

The Ankom Daisy^II^ incubator (AD^II^; Ankom Technology Corporation Fairport, NY, USA) has gained acceptance as an alternative to traditional in vitro procedures. It reduces the labour requirement and increases the number of determinations that can be completed by a single operator. The apparatus allows for the simultaneous incubation of several feedstuffs in sealed polyester bags in the same incubation vessel, which is rotated continuously at 39.5 °C. With this method, the material that disappears from the bag during incubation is considered digestible. The method, which was first developed to predict the digestibility of feedstuffs for ruminants, has been modified and adapted to improve its accuracy and prediction capacity. Modifications used by various researchers include the use of different inocula, buffer solutions, and sample weights. Recently, attempts have been made to adapt the method to determine nutrient digestibility of feedstuff in non-ruminant animals, including pets.

**Abstract:**

This review summarises the use of the Ankom Daisy^II^ incubator (AD^II^; Ankom Technology Corporation Fairport, NY, USA), as presented in studies on digestibility, and its extension to other species apart from ruminants, from its introduction until today. This technique has been modified and adapted to allow for different types of investigations to be conducted. Researchers have studied and tested different procedures, and the main sources of variation have been found to be: the inoculum source, sample size, sample preparation, and bag type. In vitro digestibility methods, applied to the AD^II^ incubator, have been reviewed, the precision and accuracy of the method using the AD^II^ incubator have been dealt with, and comparisons with other methods have been made. Moreover, some hypotheses on the possible evolutions of this technology in non-ruminants, including pets, have been described. To date, there are no standardised protocols for the collection, storage, and transportation of rumen fluid or faeces. There is also still a need to standardise the procedures for washing the bags after digestion. Moreover, some performance metrics of the instrument (such as the reliability of the rotation mechanism of the jars) still require improvement.

## 1. Introduction

The in vitro digestion method was first developed as an alternative to the costly, labour-intensive, time consuming, and ethically difficult in vivo method to predict nutrient digestibility in ruminants.

The first method, described by Tilley and Terry [1] as a two-stage rumen fluid–pepsin technique (TT), provided satisfactory estimates of in vivo apparent digestibility [2], although some authors found that the TT was just accurate for fresh grasses and not for silages or straw [3,4,5]. Van Soest et al. [6] (VS) and Goering and Van Soest [7] (GVS) modified the TT by replacing the acid–pepsin step with a neutral detergent digestion step; this version of the method is faster and more accurate than the original TT, and it is able to estimate the in vitro true digestibility of feedstuffs on the basis of the undigested cell-wall constituents.

In an attempt to overcome problems related to the variability of the rumen fluid [8], Czerkawski and Breckenridge [9] developed a continuous-culture system using an apparatus described by Gray et al. [10] and by Aafjes and Nijhof [11] as a starting point: the “RUmen SImulation TEChnique” (Rusitec), which is still successfully used to generate inocula for in vitro studies [12,13,14].

Other in vitro methods have been developed to estimate the digestibility of feedstuff. Menke and Steingass [15] proposed to measure the gas produced during fermentation and feed composition data to estimate the energy content of feeds. Theodorou et al. [16], considering previous studies [17,18], developed an in vitro method to measure the accumulation of head-space gas; this method was then revised by other authors, who used computerised pressure sensors to monitor the gaseous products of the microbial metabolism and found a clear linear relationship between the disappearance of neutral detergent fibre (NDF) and the production of gas [19,20].

The need for a piece of apparatus that would be capable of automating traditional in vitro digestibility analysis and resolving some analytical errors such as those pertaining to sample handling and manual filtration steps led to the development of the Ankom Daisy^II^ incubator (AD^II^; Ankom Technology Corporation Fairport, NY, USA).

This review summarises the use of the AD^II^ incubator—from its introduction until today—in digestibility studies on ruminants, compares and correlates it with other digestibility procedures, and discusses the sources of variability of the results and the extension of this technology to other non-ruminant species. Finally, some hypotheses on the future evolution and development of this technology and on the standardisation of the procedure are presented

## 2. The Ankom Daisy^II^ Incubator

The AD^II^ incubator started out as a project for a Canadian customer and was introduced to the public in 1994 as a wooden and somewhat fragile cabinet [21]. In 1997, a new model was made with a more resistant metal cabinet, exactly as in the currently marketed form (Figure 1). AD^II^ is essentially based on the in vivo simulation of digestion. With this device, it is possible to simultaneously analyse up to 92 samples in a thermostatically controlled chamber that contains four rotating digestion jars. The temperature inside the chamber is maintained at 39 ± 0.5 °C by a heat controller; a timer allows each incubation period to be set. Samples are weighed in F57 filter bags (25 μm pore size) (Ankom Technology Corporation Fairport, NY, USA) and put into the jars (up to 23/jar) together with the inoculum (rumen fluid, faeces, or enzymes) and a buffer solution. Each of the four glass jars, placed on the rotation racks inside the incubator, contains a perforated agitator baffle that divides the internal volume into two parts and allows for the free movement of the digestion medium. The bags are weighed before and after a specific period of incubation, and the material that has disappeared is considered digestible dry matter. The AD^II^ incubator offers advantages in time, efficiency and labour requirements over conventional methods, such as the Tilley and Terry method and the Van Soest method. Because of its design, the AD^II^ design is capable of testing a large number of samples [22,23,24]. It has been identified as an easy, inexpensive, and efficient instrument for the prediction of the digestibility of several feedstuffs and diets [8,22]. However, compared to other techniques (such as the batch culture technique, the use of the Ankom Gas Production System or the Rumen Simulation Technique), the AD^II^ incubator has been demonstrated to give higher values at different incubation times [25].

One application of the AD^II^ incubator is the estimation of neutral detergent fibre digestibility (NDFD) at single time points (such as 30 or 48 h) [26].

Attempts to address the variability of results have involved the assessment of the vessel type and the sealing, venting, and gassing procedures [27]; the comparisons of different types of fibre-bag and the use of sodium sulphite for long incubation periods [28]; the development of specific in vitro methods to determine indigested NDF and to estimate the individual pool sizes and rates of digestion for application for diet formulation purposes [29]; the evaluation of the storage times and temperatures of rumen fluid before its transfer to the incubation flask [30]; the effects of the priming techniques of rumen fluid [31,32]; comparisons with in situ and various in vitro methods [33,34]; and the quantification of two pools of digestible NDF (fast and slowly digested) with a minimal number of fermentation time points [35].

Recently, the AD^II^ incubator has been used for the in vitro long-term ruminal digestion (240 h) of undigested NDF (uNDF) [28]. To estimate the kinetics of NDF degradation, longer time intervals are essential, especially when using complex models. Complex models may require inputs of fast, slow, and indigestible NDF pools [35,36], which can be determined with ease when using the AD^II^ incubator.

A list of practical recommendations on the use of AD^II^ incubator and a list of the main problems concerning the use of the instrument that require further study are reported in Appendix A.

## 3. Inoculum Applied to the Use of the Ankom Daisy^II^ Incubator

Inoculum is very important for in vitro fermentation studies, but it also represents the greatest source of uncontrolled variation in fermentation systems. The inoculum has to create a similar environment to that of the digestive tract [37], but its digestive capacity may be influenced by the animal species, breed, and individual and within animal variations from time to time [38]. The characteristics and quality of the inoculum is not a specific problem of the AD^II^ incubator. As there is a lack of specific information on the AD^II^ incubator and some authors have studied inoculum for in vitro analysis, we reported their experience with other digestibility systems, because this information may also be useful for Ankom Daisy^II^.

### 3.1. Rumen Fluid

As for other systems, the most frequently used inoculum source in the AD^II^ incubator is rumen fluid (RF). The necessity of fistulated and cannulated animals to provide this inoculum raises a number of practical problems, e.g., the need for surgical facilities, constant care to avoid infections, and the costs associated with the long-term maintenance of these animals. Moreover, the use of cannulated animals for this purpose has been criticised on ethical grounds.

Different solutions, in which there is no need to use cannulated animals, have been studied to resolve cost issues and ethical concerns about the well-being of animals. RF can be obtained via the oesophagus, thereby avoiding the need for cannulation, but such samples are often contaminated with saliva, and their collection causes considerable stress to the host animal. Moreover, as a result of the placement of the sampling device, the samples may not be representative of the entire rumen contents [37]. A very different approach with more details on this matter can be found in a paper by Ramos-Morales et al. [39]. These authors assessed in vivo trials conducted with ruminally cannulated sheep and goats to validate the use of stomach probing as an alternative to rumen cannulation in small ruminants with the aim of detecting any differences in ruminal fermentation and in the microbial community between species, diets, and sampling times.

A more ethically acceptable approach that reduces stress and alleviates the suffering of animals by avoiding an invasive procedure is the collection of RF at slaughtering [40]. Alba et al. [41] verified, through the use of an AD^II^ incubator, that the rumen inoculum obtained from slaughtered cattle can be used to replace the use of cannulated animals and that this approach is a viable alternative to digestibility analysis.

This method is accepted by the Rumen Microbial Genomics Network [42] for microbiota studies and has been mentioned as an alternative to sampling via cannula [43].

A supplemental video of the sampling procedures of RF at slaughtering is available online [44]. These procedures involve the collection of the rumen content into plastic bags a few minutes after slaughtering; the rumen content is squeezed, and the RF is filtered and collected into pre-heated plastic bottles. The presence of oxygen is avoided by squeezing the bottles while closing them; the rumen fluid is transported to the lab (max. 1.5 h time) at a temperature of 39–42 °C.

The effects of the source of inoculum with various combinations of donor cow diets generally vary to a great extent [45]. The results of a trial conducted by Holden et al. [22] showed that the source of inoculum affected in vitro dry matter digestibility (DMD). A grass hay donor cow diet resulted in lower digestibility values than a corn silage-based, total mixed ration donor cow diet for alfalfa hay, grass hay, steam flaked corn, and dry ground corn. No influence of the donor diet was found for mixed haylage, corn silage, grain mixture, or high moisture shelled corn. King and Plaizier [46] found that the source of inoculum (steers or cows) did not affect apparent or true DMD to any great extent. They also found that forage digestibility was similar when using the RF from sheep and from cattle [23]. Ammar et al. [47], using an AD^II^ device, found that the RF of sheep and goats was similar under the conditions of the experiment when all the donor animals were fed the same diet and were maintained under the same conditions.

Robinson et al. [30] examined the influence of storage time and temperature on the ability of rumen microorganism to degrade NDF. They reported that within-day delays of up to 6.5 h between the time of collection of rumen inoculum and the time of the initiation of the in vitro incubation had no impact on the measured 48 h digestion of NDF if the RF was maintained at 39 °C under anaerobic conditions during the delay. Similarly, the RF of sheep, preserved for up to 6 h in crushed ice, had no effect on any fermentation parameters [48]. Another possible RF storage system for in vitro incubation is short-term refrigeration [41]. Chaudhry and Mohamed [49] tested thawed RF from frozen rumen contents (stored at −20 °C for 4 w) against fresh RF from the same slaughtered cattle. Though the thawed RF had a lower degradation than the fresh one, it could be used to predict in vitro digestibility, as the values were closely correlated (R^2^ = 0.95). However, it was still necessary to test its suitability for routine use. Hervas et al. [48] instead found a reduction in fermentative activity as a result of freezing (24 h). Spanghero et al. [14] recently compared inoculum collected at slaughtering with RF samples obtained from a continuous fermenter that were fresh, refrigerated at 4 °C, chilled at −80 °C, and freeze-dried. They evaluated the fermentability by measuring the NDF, crude protein degradability, and gas production. They confirmed that short-term refrigeration is a valuable technique to manage RF, whereas methods based on low temperatures significantly reduce the *Fibrobacter succinogenes,* which are very important for fibre degradation. Denek et al. [50] studied the preservation of microorganisms with a cryoprotectant under different deep-frozen conditions. They showed that RF treated with 5% dimethyl sulphoxide and frozen in liquid nitrogen gave similar results to fresh RF, but they also showed that the incubation time needed to be increased to 72 h to measure the digestibility of roughages. Belanche et al. [51] assessed the relevance of different factors (the diet of the donor animal, the fermentation substrate, microbial fraction, and the inoculum preservation method) to maximize the rumen inoculum activity, and they found that the highest microbial numbers and in vitro fermentation rates were recorded for fresh RF sampled after 3 h from donor animals fed a high concentrate diet.

As far as the microbial population that develops in an AD^II^ incubator is concerned, Soto et al. [52] showed the variations such a population underwent during the incubation process, and they compared the results with those of a Wheaton bottle and a single-flow continuous-culture fermenters using the same goat RF. In an AD^II^ incubator, they monitored the different microbial groups (bacteria, archaea, fungi, and protozoa) for 48 h by means of real time-PCR and terminal-restriction fragment length polymorphism. They observed a general decrease in the microbial population and important changes in microbiota profile, as the methanogens population increased. A similar trend was observed for the Wheaton bottle at 72 h, but there was also a growth of fibrolytic bacteria. However, the continuous-culture fermenters kept the rumen microbiota similar to that sampled from the rumen.

Spanghero et al. [14] found that the fermentation liquid from rumen continuous-fermenters can be used to generate inoculum for in vitro purposes.

Problems can arise for microorganisms, regarding the preparation of inoculum [37], connected with feed particles, the use of multiple layers of cheese cloth, and/or the use of some physical methods (e.g., the Stomacher method or the maceration of the rumen content in a food processor), which may destroy cell integrity.

### 3.2. Faecal Inocula

Fresh faeces (FF) have been used as an alternative source of ruminal inoculum in many experiments [41]. All these studies have demonstrated that bovine faeces may be used as microbial inocula for in vitro digestion and gas production, but this use has some limitations, such as a lower enzymatic activity than RF [53,54,55]. According to Akhter et al. [56], cattle faeces could also be used as an alternative to sheep RF.

Tufarelli et al. [57] tested faecal samples of yaks (*Bos grunniens*) as an alternative microbial inoculum source and compared them with RF, which was used as a control. They found that a faecal extract could be utilised instead of RF to estimate in vitro digestibility and that an AD^II^ incubator, with faecal liquor, is able to simply assess the adaptation capability of ruminant species to a pasture. These results were confirmed using camel faeces as a source of inoculum for AD^II^ [58].

Bovine FF may be used to replace bovine RF for incubation times no lower than 48 h [59]. Chiaravalli et al. [60] utilised an AD^II^ incubator to estimate the undigestible NDF of seven substrates using three different inocula (one rumen and two faeces) and considering two incubation times (240 and 360 h). The undigestible NDF results showed that faecal inoculum could be used to replace RF for long incubation times and that faeces can be used as an inoculum for end-point measurements.

The diet of an animal can change its microbial population. Guzmán and Sager [61] compared the microbial inoculum collected from a rumen-fistulated Aberdeen Angus steer fed with alfalfa hay and then with low quality digit hay (*Digitaria eriantha*), as well as the faeces collected from the same animal to evaluate the substrate, inoculum, and digestibility interaction. Using both inoculum sources, the true DMD was found to be affected by the diet of the donor animal, and the RF values ranked higher in the runs. Moreover, Kim et al. [62] suggested considering the diet, because it has an important effect on faecal microbiota, in particular when a forage-based diet is compared with a concentrate.

Faeces have also been extensively used as inoculum for in vitro incubation trials on monogastrics. Lowman et al. [63] were the first to demonstrate that equine faeces can be used as a source of microbial inoculum and that the faecal microflora of equines can remain viable for several hours after excretion. Other authors have confirmed these results. Earing et al. [64] demonstrated that the in vitro methodologies developed for the AD^II^ incubator could produce accurate estimates of in vivo equine apparent DMD and NDFD when equine faeces were used as the inoculum source. They evaluated three incubation periods in their study: 30, 48, and 72 h. Though the 30 and 48 h in vitro estimates were consistently less accurate than the in vivo estimates, they ranked diets in the same order as the in vivo method, and the 72-h period provided the most similar digestibility estimates to the in vivo data. Tassone et al. [65] evaluated the use of the AD^II^ incubator for the apparent and true DMD and NDFD measurements of feedstuffs considering four incubation times (30, 48, 60, and 72 h) using donkey faeces as a source of microbial inoculum. All the digestibility parameters increased significantly after 30–72 h of incubation, with average coefficients of variation for repeatability and reproducibility of 3.4% and 7.3% for apparent DMD; 1.7% and 4.3% for true DMD; and 6.6% and 14.6% for NDFD, respectively.

Table 1 summarises the references pertaining to rumen fluid and fresh faeces inocula applied to the AD^II^ incubator.

### 3.3. Enzymatic Inoculum

Enzymatic methodologies, in which microbial inoculum is eliminated, were developed to avoid problems associated with variations in rumen fluid over time [37]. This approach can be recommended because it offers an improved standardisation of the methodology, a reduction in the variations that may be attributed to the inoculum source and preparation, and a reduced dependence on surgically modified animals as rumen fluid donors [66]. However, the attempt to use enzymes instead of rumen fluid or other inocula have resulted in problems of variability in their preparation [67], and very little work has been done to optimise enzyme activities or incubation conditions. Though there are no available studies on ruminant digestibility in which enzymes were used in an AD^II^ incubator, many authors have already used enzymes in digestibility studies on pigs [68], rabbits [69,70], and dogs [71].

## 4. Sample Size, Sample Weight and Bag Type

The sample bags in the AD^II^ incubator constantly rotate in jars (0.95 rpm), and the internal septum leads to the complete immersion of the bags at every spin of the jar; in this way, gases do not accumulate inside the bag, and samples are prevented from floating freely in the flask. The continuous shaking of samples produced significantly higher digestibility results than when shaking occurred only twice daily [72]. As reported by Alende et al. [25], the use of filter bags may be advantageous, because filtration and recovery have been mentioned as sources of variability of the digestibility coefficients. Additionally, jars positioned horizontally render a higher digestibility than vertically placed ones. Holden et al. [22] found no significant differences when grains and forages were incubated in the same digestion vessel.

The first and most extensively used AD^II^ incubator bag is the F57 bag. The F57 bag is made up of an extruded polyethylene fibre with a three-dimensional filtration matrix that facilitates the maximum flow of a solution, thereby obtaining the best substrate interaction and minimum particle loss. The F57 filter bag has an approximately 25 μm pore size, is 50 mm long and 50 mm wide at the open top, and tapers to a bottom width of 30 mm. Sample processing, particularly concerning the grind size, interacts with the pore size of the bag and affects the extent of feed disappearance [73]. The ratio of the sample size to the bag surface area, suggested by Vanzant et al. [74] to increase the accuracy of degradability predictions relative to in vivo ruminal disappearance, is 10 mg/cm^2^.

In previous studies, sample sizes of both 0.25 g [28,30] and 0.5 g [33,34] were used in conjunction with Ankom procedures [75]. Coblentz and Akins [76] compared the NDF digestibility values of triticale forages determined with the AD^II^ device, and they considered two sample sizes (0.25 and 0.50 g) and incubation periods of 12, 24, 30, 48, 144, and 240 h. The results were compared with those obtained from a commercial laboratory that used a traditional methodology. With the 0.25 g sample size, the linear equations between the Ankom and the traditional methods did not show differences both 30 and 48 h. There was less agreement, particularly for the 30 h incubation, when a sample of 0.50 g sample was used. The NDF digestibility values were generally greater for the 0.25 g sample size when using the Ankom methods, especially for incubation times of 24, 30, and 48 h.

Cattani et al. [77] evaluated what sample size (0.25 or 0.50 g/bag) allowed for a better correlation to be achieved between the NDFD and true DMD values obtained with the AD^II^ and a conventional batch culture technique. The regressions between the mean values, provided for the various feeds by the two methods, for the NDF and true DMD, had R^2^ values of 0.75 and of 0.92,and an RSD (relative standard deviation) of 10.9% and of 4.8%, respectively, for the 0.50 g/bag size. The corresponding regressions for NDFD and true DMD showed R^2^ values of 0.94 and of 0.98 and an RSD of 3.0% and of 1.3%, respectively, for the 0.25 g/bag size. This screening analysis therefore indicated that the reduction of the sample size from 0.50 to 0.25 g of feed sample/bag (corresponding to 12 and 6 mg/cm^2^ of bag surface), when using an AD^II^ device, allowed for more closely correlated and less variable estimates of NDFD and true DMD to be obtained than those provided by the batch culture technique.

A recent work that evaluated the rate kinetics of triticale forages considered 0.3 g samples sealed within fibre bags as a procedural compromise between the 0.25 g sample size recommended for short incubation times and the necessity of ensuring that an adequate amount of residue remained after a long digestion time (144 and 240 h) [78].

The critics of the Ankom bag method have indicated the potential loss of small indigestible particles through its pores and that any method should decrease the loss of small particles without restricting access to the protozoa and bacterial populations. Ankom recommends F58 for crude fibre, neutral, and acid detergent fibre analyses. A pore sizes of <10 μm can restrict the number of protozoa and bacteria that enter digestion bags, so a smaller bag pore size than that of F58 is not advisable. Wilman and Adesogan [23] verified that soluble matter from samples high in soluble substances is able to escape from F57, thereby influencing the microbial population and increasing cell wall degradation in any samples low in soluble substances that are in the same jar. Valentine et al. [28] compared Ankom F57 bags (25 μm) with F58 bags (8–10 μm pore size) to measure undigested NDF after 240 h of incubation and found that both had significant effects on lowering undegraded NDFom values. In conventional procedures, smaller pore size filters generally tend to have greater average undegraded NDFom values than methods with larger pore size filters. They expected a similar finding, because potentially undigested NDF may be retained by finer filters, whereas potentially indigestible and digestible NDF may inadvertently escape from a coarser filter. They found when using the same technique for in vitro analysis, that Ankom F57 and F58 gave similar digestion rate results.

Adesogan [79] tested alternative bags to Ankom F57. He determined the in vitro apparent dry matter digestibility of the feed samples in an AD^II^ incubator using Ankom F57 bags and dacron bags with pore sizes of 30 and 50 μm, with or without a 5 g glass ball placed in the bags to ensure submersion in the media. He obtained different digestibility estimates when the alternative bags were used instead of the F57 bags, but the Ankom bags gave a more precise prediction of conventionally measured digestibility estimates than the alternative bags. Using Ankom bags ensures more standardised and repeatable results. The characteristics of alternative bags should be disclosed whenever they are used, instead of F57 bags, to estimate digestibility. Anassori et al. [80] also used dacron bags (pore size of 50 µm) in an AD^II^ to measure the organic matter digestibility (OMD) of forage-based sheep diets supplemented with raw garlic, garlic oil, and monensin. They compared AD^II^ with the TT and gas production. The values obtained with the AD^II^ method were always higher than those obtained with the TT and (for diets containing garlic oil) with in vitro gas production methods. According to the authors, in the AD^II^ procedure, a proportion of non-digestible fine particles may have been removed during incubation, boiling, and rinsing, thus reducing the weight of the residue and increasing the estimate of digestibility compared to that obtained with other methods.

Table 2 summarises the references pertaining to the sample size and bag type applied to the AD^II^ incubator.

## 5. Buffer Solutions and in Vitro Digestibility Methods Applied to the Ankom Daisy^II^ Incubator

Many methods and buffer solutions that are used to study in vitro digestibility, first for ruminants and then for monogastrics, have also been applied to the AD^II^ incubator.

A buffer solution (either phosphate, carbonate, or both) is used during incubation to control the pH and to supply nutrients for the inoculum microorganisms. Without a buffer, the short chain of fatty acids would lower the pH [81]. As authors have reported, only phosphate buffers do not require preparation under CO_2_. The references of the different buffer solutions used for in vitro digestibility analysis are briefly reported in Table 3. However, a comparison of buffer solutions is still lacking. In 2000, Figueiredo et al. [72] compared buffers that had been described by Marten and Barnes [82] with those that had been described by Minson and McLeod [83], and the authors verified that the solutions could replace each other.

## 6. Precision and Accuracy of the Method Using the Ankom Daisy^II^ Incubator

The utilisation and the diffusion of AD^II^ to study in vitro digestibility is a result of the reliability and accuracy of the method.

Damiran et al. [87] found a coefficient of variation (CV) of 4.7% for DMD measured with AD^II^ and a CV of 12.2% for NDFD. A CV of <1% was observed between sample replicates in other laboratories for the in vitro true digestibility values, but this coefficient normally ranged between 1–3% [21]. However, it is a little higher for NDFD analysis and typically ranges from 2.0–4.5%, depending on the type. Corn silage samples are always a little more variable. If any sample has a CV of over 5%, it should be re-analysed. Figueiredo et al. [72] verified a good reproducibility when measuring digestibility with AD^II^. They reported a low coefficient of variation (CV = 2.65%) between jars and within jars, with values of 3.92, 2.13., 6.12, and 1.94 for jar numbers of 1, 2, 3, and 4, respectively. Tagliapietra et al. [88], in situ and in vitro, studied the rumen fluid of 11 feeds collected by means of oro-ruminal suction from intact donor cows. The reproducibility coefficient of the DMD for AD^II^ was 96.0%. The DMD values were underestimated when filter bags were considered, compared to in situ-nylon bags and in vitro conventional bottles. Nevertheless, it was possible to overcome the lower repeatability provided by the filter bags by increasing the number of replicates: three filter bags led to approximately the same standard error as the mean of 2.5 nylon bags and the mean of 2 conventional bottle measurements. The results showed a direct proportionality between the DMD values obtained in situ and in vitro with different techniques (in situ nylon vs. in vitro conventional bottles and in situ synthetic filter bags vs. AD^II^).

Spanghero et al. [89] studied the NDF degradability of 18 hays considering different incubation times (2, 4, 8, 16, 24, 48, and 72 h) and found that the variability (CV) of the AD^II^ incubator (including jar repeatability) was 2.8%—that is, a similar value to that generally found for some chemical analyses of feedstuffs [90] and one that is lower than that obtained for in situ measurements (including low repeatability, CV: 3.7%).

Spanghero et al. [91] also evaluated the precision of the AD^II^ device in measuring the in vitro NDF degradability of 162 hay samples from permanent Austrian grasslands. The obtained results showed a within forage standard error of 2.8%. This limited repeatability of the measurement was attributed to various sources of variability (bag porosity, dimensions, amount of substrate, etc.), but not to the different jar positions in the fermenter, because the average values obtained after five incubations for the different jars were not statistically different.

Spanghero et al. [92] also investigated the precision and accuracy of the AD^II^ incubator for NDFD analysis and the accuracy and reproducibility of the associated calculated net energy of lactation. Five laboratories analysed 10 fibrous feed samples each; the fermentation times in the AD^II^ incubator were 30 and 48 h. The precision was measured as the standard deviation (SD) of the reproducibility (SR) and repeatability (Sr) of the between and within laboratory variability. Extending the fermentation time from 30 to 48 h increased the NDFD values (from 42% to 54%) and improved the NDFD precision, in terms of both Sr (12% and 7% for 30 and 48 h, respectively) and SR (17% and 10% for 30 and 48 h, respectively). The 48-h period of incubation improved the accuracy and reproducibility of the calculated net energy of lactation.

The accuracy and precision of NDFD, determined after short or long-time intervals, has recently been of considerable research and industry interest, as the relative consistency of the results.

Cişmileanu and Toma [93] studied the repeatability, reproducibility, and accuracy of an AD^II^ incubator using a new version of the TT. The stages of the method were similar to those of the traditional version: one stage with buffered rumen liquid and one stage with pepsin–HCl. An alfalfa hay sample was tested to establish the OMD by means of the in vivo method, and it was then considered as an internal control feed with a known digestibility. The authors observed that the coefficient of variability was 1.11% for repeatability and 1.85% for reproducibility. The accuracy was the same as that obtained with the conventional method.

Moreover, even if the AD^II^ incubator is fully functional, sometimes the jars do not rotate correctly and suffer from slowdowns, stops, and starts [94]. Some structural adjustments are therefore necessary to better exploit the potential of the AD^II^ incubator and to implement its diffusion and use.

Table 4 summarises the references pertaining to the precision and accuracy of the method using the Ankom Daisy^II^ incubator.

## 7. Comparison with other Methods

Many methods are available to measure in vitro digestibility, but only a few articles have compared the results obtained using an AD^II^ incubator with the results of other procedures [25].

The first results on digestibility in ruminants obtained using an AD^II^ incubator were presented by Komarek et al. [95] in 1994 at the National Conference on forage quality in Lincoln (USA) [96]. The following year, Ayangbile et al. [97] showed that there were no differences between DMD data obtained from an AD^II^ incubator and data obtained by means of the conventional Tilley and Terry methods [1,7]. Traxler et al. [98] determined the true DMD on four forages for different incubation times (48, 72, and 144 h), and even though the conventional Van Soest method [6] was found to be more efficient, the results basically confirmed the conclusions of Ayangbile et al. [97].

Cohen et al. [99] incubated corn silage samples in tubes according to the GVS method [7] and in an AD^II^ incubator at different times using unwashed F57 bags or F57 bags washed in acetone before being filled. The NDFD measured with the AD^II^ incubator was lower than that in the tubes, probably because of the retention of gas and acid end products within the bags, and the values of the washed filter bags were similar to those obtained by shaking the tubes. Traxler [100] instead noted very few differences between the AD^II^ incubator and the GVS method [7].

Over time, other studies have confirmed that the AD^II^ incubator can be used to predict the DMD digestibility of forages, grains, and mixed rations for ruminants [7,22,23,24,26,73,87,101].

Ammar et al. [102] compared the TT and VS methods [6] using an AD^II^ incubator for leguminous shrub species. The medium was prepared according to the VS method. After incubation in a buffered rumen fluid, samples were either subjected to a 48 h pepsin–HCl digestion (TT) or gently rinsed and extracted with a neutral detergent solution at 100 °C, as described in the VS method. The apparent digestibility was generally lower than the true digestibility, and the differences were always significant, particularly in leaves.

The same author [103] used the VS method applied to the Ankom technique [104] to obtain the in vitro digestibility of the stems and leaves of grasses and legumes taken from the first and subsequent cuts of a permanent meadow. In this experiment, rumen fluid was withdrawn from adult sheep.

Gargallo et al. [85] verified the use of an AD^II^ incubator to determine the intestinal digestion of crude protein using Calsamiglia and Stern’s three-step procedure (TSP) [84]. Four tests were conducted to study the effect of the type of pepsin, the type of bags, the amount of sample, and the number of bags per jar on the estimated intestinal digestion using the AD^II^ incubator and the TSP techniques on soybean meal samples, heated at different temperatures, and with 12 protein supplements. The results showed that the intestinal digestion of soybean meal and the 12 protein supplements from the TSP and the AD^II^ incubator (with R510) were closely correlated. The amount of sample per bag and the number of bags per jar did not affect the estimates, and up to 30 bags (Ankom R510) with 5 g of sample could be used in each jar of an AD^II^ incubator to estimate the intestinal digestion of the proteins in ruminants.

In 2017, Cişmileanu and Toma [93] successfully validated a new version of the TT applied to AD^II^, in which the stages of the traditional procedure were maintained. Two stages, the first one with buffered rumen liquid and the second with the pepsin–HCl solution, were considered.

Holden et al. [22] compared a modification of the TT and the AD^II^ incubator techniques to determine DMD, considering sources of inoculum from two different donor cow diets, as well as all the forage and total mixed rations. Their results showed that the AD^II^ incubator did not affect the digestibility values of the forages or grains to any great extent, as well as that the source of inoculum could affect DMD.

Wilman and Adesogan [23] compared the TT and an the AD^II^ incubator to estimate apparent and true DMD, apparent and true OMD, and NDFD. The analysed forage samples comprised 72 combinations of two forage species (*Lolium multiflorum* and *Medicago sativa*), three plant parts, three degrees of particle breakdown, two field replicates with rumen fluid from sheep, and two field replicates with rumen fluid from cattle. It was found that the sieve size used when milling did not influence the true OMD. However, small differences were observed between the two forage species: the standard errors and coefficients of variation were higher for the AD^II^ incubator (mean: 4.0%) than for the TT (mean: 2.7%). When they used the TT, they found it was possible to more precisely predict the true digestibility than the apparent digestibility from the AD^II^ incubator results; the difference between apparent and true digestibility, when estimated using the AD^II^ incubator, appeared unrealistically low. The estimated digestibility was similar when rumen fluid from sheep and from cattle was used. In conclusion, the TT gives more precise results than the AD^II^ incubator, albeit at the cost of requiring more labour. Mabjeesh et al. [73] performed the same comparison (AD^II^ vs. TT) on 17 concentrates and protein supplements, and they obtained a satisfactory relationship (R^2^ = 0.81), even though the AD^II^ incubator gave higher values for some energy concentrate and protein supplements.

Ricci et al. [105] compared the precision and accuracy of in vitro ruminal DM degradability using the TT, an AD^II^ incubator, and the gas-production technique to estimate the in vivo DM digestibility of tall wheatgrass, hay, and haylage. The goodness-of-fit of all the techniques with the in vivo DM digestibility and the relationships between them were evaluated by means of a simple linear regression analysis. The Pearson correlation coefficient (ρ) was used to evaluate the strength of the association between the observed and in vitro estimated data. The concordance correlation coefficient (ρc) was used as a single indicator to integrate both precision and accuracy (Cb). This indicator (scaled between 0 and 1) is a reproducibility index that evaluates the agreement between two sets of data by measuring the shift in location from the concordance line (the 45° line through the origin) in the observed versus predicted plot. Cb is a bias correction factor that indicates how far the best fit line deviates from the concordance line. Linear relationships were observed between the in vivo and the TT, AD^II^, and gas production values. The TT had the highest correlation (0.98), and this was followed by the gas-production technique (0.97) and then by AD^II^ (0.96). However, the TT exhibited the lowest accuracy (ρc = 0.341), and AD^II^ exhibited the highest (ρc = 0.850). The regression analysis showed an overestimation of the in vivo dry matter digestibility above 48.8% for AD^II^ and an underestimation below this value. AD^II^ is faster and more accurate than the other techniques, and it therefore appears to be the most suitable for in vitro digestion trials. Figueiredo et al. [72] compared the AD^II^ technique with Minson and McLeod’s technique [83], (they modified the TT in 1972) and found higher values when they used the AD^II^ procedure.

Some authors have conducted comparison between an AD^II^ incubator and in situ system. Robinson et al. [30] reported higher NDFD values at 48 h with an AD^II^ incubator. Spanghero et al. [92] showed that the results of an AD^II^ incubator were closely correlated with the results of an in situ method (R^2^ = 0.98). Spanghero et al. [89] compared the NDF degradability of 18 hays, measured by means of an in situ method (nylon bag technique) and the AD^II^ incubator. The incubation times were 2, 4, 8, 16, 24, 48, and 72 h. The NDFD values obtained in situ and in vitro with the AD^II^ incubator after 48 h of incubation were closely correlated (R^2^ = 0.94). In another study [91], they verified that the NDF degradability of 162 hay samples measured in an AD^II^ incubator was 25–30% higher than the effective in situ values. The regression analysis between the in vitro and in situ NDFD values showed a medium degree of correlation and a low level of accuracy.

Tagliapietra et al. [88] compared four in situ methods with nylon bags and filter bags, as well as in vitro with conventional individual bottles or AD^II^, to measure the DMD of 11 feeds. The reproducibility coefficients of the dry matter digestibility were 97.9%, 95.1%, 98.8%, and 96.0% for the in situ-nylon, filter bags, conventional bottles, and AD^II^, respectively. The in situ and in vitro filter bags underestimated the dry matter digestibility values compared to the in situ-nylon bags and conventional bottles. They concluded that in vitro estimates of dry matter digestibility at 48 h with AD^II^, using rumen fluid collected from intact cows, can produce similar values to those obtained in situ. The filter bags underestimated the dry matter digestibility values compared to the in situ-nylon bags and conventional bottles. However, it was possible to overcome the lower repeatability provided by the filter bags by increasing the number of replicates: three filter bags gave approximately the same standard error as the mean of 2.5 nylon bags and the mean of two CB measurements. The results showed a direct proportionality between the dry matter digestibility values obtained in situ and in vitro with different techniques (in situ-nylon vs. conventional bottles and in situ-filter vs. AD^II^).

Alende et al. [25] compared three different DMD methods (AD^II^ incubator, batch culture, and Ankom gas production) considering four incubation times (12, 24, 36, and 48 h); the results obtained at 24 h were compared with those obtained from dual-flow, continuous-culture fermenters. The results showed that different methods yield different DMD values. When the incubation time was longer than 12 h, the predicted DMD from the AD^II^ incubator was greater than when the gas production and the batch culture methods were used. The apparent DM digestibility, estimated using the continuous culture fermenter, was similar to that obtained from the batch culture and gas production, but it was lower than that of the AD^II^ incubator. Damiran et al. [87] concluded that the AD^II^ technique is able to accurately predict in vivo and the in situ DMD. Table 5 summarises the references pertaining to comparisons with other methods.

## 8. Use of Daisy^II^ Incubator for Non-Ruminants

### 8.1. Horses

The in vivo standard and the inert marker methods are optimal for the determination and assessment of the digestibility of horse feeds, but they are time consuming. The use of in vitro fermentation procedures, such as enzyme-based essays, for the prediction of pre-caecal starch digestibility [106], and the gas production technique, developed for ruminants [15] using either caecal fluid [107] or faeces as inocula [108] to study diet digestion and fermentative end products has become increasingly more popular in equine nutrition. Abdouli and Attia [109] developed a simple in vitro method that is suitable for both concentrates and forages and that combines both the pre-caecal and hind gut digestion processes. These authors focused on the duration needed to establish feed pre-digestion by pepsin–amylase and its subsequent effect on gas production and organic matter digestibility using horse faeces as a source of microbial inoculum, and they compared the results with those from low-to-high-starch and protein feeds. They concluded that this procedure should be extended and validated with a large array of feeds with known digestibility values, because the enzymatic pre-digestion treatment effects varied between samples (non-pre-digested hay, barley grain, and soybean meal). Equine faeces is a suitable source of microbial inoculum for in vitro gas production studies, and the evaluated in vitro batch culture technique showed a considerable potential for the routine prediction of the nutritive value of a wide range of equine feedstuffs [79].

Lattimer et al. [110,111] studied the effects of *Saccharomyces cerevisiae* on the in vitro fermentation of a high concentrate or high-fibre diet for horses using equine faeces as an in vitro inoculum source in an AD^II^ incubator. These authors demonstrated that the use of 0.25-g samples may yield more accurate and less varied estimates of DM digestibility. Furthermore, the DM digestibility values for the in vivo and in vitro were similar, and they concluded that the AD^II^ incubator could be used to predict the DM digestibility of diets. Earing et al. [64], evaluating the in vitro digestion of four different diets using the AD^II^ incubator, recently confirmed that equine faeces are a suitable source of microbial inoculum for in vitro digestibility studies on horses. They found comparable DM digestibility for diets consisting of timothy hay, timothy hay with oats, and alfalfa hay with oats between in vitro and in vivo methods, while different digestibility values were observed between the two methods for an alfalfa hay diet. These authors stated that further research is needed, using a wider range of forages and methods, to determine whether in vitro and in vivo digestibility methods produce similar results for horses and to establish in vitro digestibility as a viable technique for estimating digestibility in horses.

Blažková et al. [112] compared the in vivo DM digestibility of corn silage for horses with that obtained using equine faeces in an AD^II^ incubator. These authors concluded that DM digestibility is only comparable with data on ruminants, and they showed that horses have a lower DM digestibility of corn silage than ruminants. Moreover, they demonstrated that equine faeces are a suitable source of microbial inoculum for in vitro digestibility.

### 8.2. Donkeys

Despite the increasing interest in donkeys, studies on this species are very limited. Tassone et al. [65] demonstrated that donkey digestibility can be predicted, with a high repeatability and reproducibility, using an AD^II^ incubator, a closed-system fermentation apparatus, and donkey faeces as a source of microbial inoculum. Moreover, these authors observed that the digestibility of different feeds for donkeys needs different incubation times.

### 8.3. Camelids

In vitro TTs that use camel rumen liquor as an inoculum require fistulated animals to provide this inoculum [113,114]. Rumen fluid can also be obtained, for the same purpose, from slaughtered dromedaries. Lifa et al. [115] therefore investigated the suitability of this rumen fluid with the aim of evaluating the in vitro degradation characteristics of highly fermentable industrial by-products (citrus, tomato, and apple), fibrous forages, and their mixtures. They concluded that rumen fluid extracted from slaughtered dromedaries is a valuable tool for determining the in vitro degradation of camel feeds. None of these experiments on camelids were conducted using an AD^II^ incubator.

The successful use of a liquid suspension of camel faeces, as an alternative inoculum for an in vitro AD^II^ incubator, yielded valid in vitro estimates of the DM, NDF, and ADF (acid detergent fibre) digestibility of forages and grains and could make it unnecessary to resort to fistulated animals (particularly in tropical countries) to obtain inoculum; this could solve some practical problems, such as the constant care needed to avoid infections and the costs associated with the long-term maintenance of donor animals, as well as ethical considerations and the necessity of surgical facilities [58].

Laudadio et al. [58] evaluated the in vitro digestibility of the fodder species browsed by camels in pastures in an arid region of Southern Tunisia using an AD^II^ incubator. They used different sources of faecal liquor, collected from camels, healthy mature sheep, and goats, as alternative microbial inoculum sources to test the nutrient digestibility of these forages, as well as rumen liquor, collected from sheep, as a control for the in vitro AD^II^ incubator. These authors stated that the similarity of the different repetitions for all the fodders in the estimation of nutrient digestibility in the AD^II^ incubator reflects its accuracy, making it comparable with traditional methods in regard to digestibility. They concluded that the AD^II^ incubator is appropriate for the determination of the in vitro digestibility of nutrients when using camel faecal liquor, which could be used instead of rumen fluid to estimate the in vitro digestibility of forages.

### 8.4. Rabbits

An AD^II^ incubator was also used in rabbit studies to determine the in vitro insoluble fibre [116] and in vitro digestibility of rabbit feedstuffs [69,70,117,118,119,120]. Abad et al. [69] adapted the in vitro digestion procedure proposed by Carabaño et al. [121] and compared the quantifications of soluble fibre in rabbit feedstuffs using different chemical and in vitro approaches. The method was modified using Ankom filter bags, which were placed in an AD^II^ incubator jar rather than in crucibles (reference method) to facilitate sample filtering. No difference was observed when crucibles and Ankom bags were used (both in single or collective digestion) for two-step pepsin/pancreatin in vitro DM digestibility, corrected for ash and protein. The correlations obtained for in vitro DM digestibility were higher (0.99) than those reported by Vogel et al. [24], who studied the in vitro DM digestibility of forages for ruminants (0.92). The latter authors reported higher in vitro digestibility when using Ankom bags than when using crucibles (0.602 vs. 0.563, respectively), whereas Abad et al. [69] found much less of a difference.

Ferreira et al. [70], in order to evaluate the potential use of dried or autoclaved sugarcane bagasse and enriched or non-enriched with vinasse in the diets of growing rabbits and to determine their in vitro dry matter digestibility, modified the last step of the Abad et al. method [69] using a caecal contents diluted at a ratio of 1:1 (w/v) with a buffered mineral solution [122] as inoculum. Ferreira et al. used the same method to determine the in vitro dry matter digestibility of rabbit diets supplemented with macaúba seed cake meal [117] or with tropical ingredients, co-products, and by-products [118].

The Ramos et al. method [123], which is based on that of Boisen et al. [124], in which Ankom bags are used, and which, in turn, was modified by Abad et al. [69], was used to determine the in vitro dry matter digestibility of rabbit diets supplemented with co-products derived from olive cake [119] or with citrus co-products [120].

### 8.5. Guinea Pigs

López et al. [125] used an AD^II^ incubator to compare two types of “in vitro” digestibility assays, using commercial enzymes and guinea pig caecal liquor with the in vivo assay to identify the assay that resembled the in vivo response the most, and they found that the optimal in vitro method to use for comparisons with the in vivo test is the caecal liquor technique because it presents a smaller difference in results.

### 8.6. Pigs

Several in vitro feed digestibility estimation methods have been developed and can be divided into three groups, that is single-, two-, or three-step models that simulate gastric digestion, gastric/small intestinal digestion, and gastric/small intestinal/large intestinal digestion, respectively [126]. The Boisen and Fernandez [127] in vitro gastric-ileal digestion procedure was been adapted for use in an AD^II^ incubator and it allows for the simultaneous incubation of different pig feedstuffs in sealed polyester bags (5 × 10 cm bags; R510, Ankom Technology, Macedon, NY) in the same incubation vessel [68].

Fushai [128] determined, with an AD^II^ incubator, the in vitro digestibility of growing pig diets supplemented with exogenous enzymes. Each feed was digested in pepsin, followed by pancreatin, with the recovery of the fibrous residues. The pepsin–pancreatin fibre extracts were digested, by means of Viscozyme and Roxazyme, in a third step to complete the simulated pig gastro-intestinal digestion process.

Torres-Pitarch et al. [129] determined the in vitro ileal digestibility of pig diets by means of a two-step in vitro incubation procedure, adapted from that of Akinsola [68] using an AD^II^ incubator at 39 °C with samples incubated inside Ankom F57 bags. The first step, which simulated the digestion in the stomach, was that of enzymatic hydrolysis with a pepsin solution at pH 2.0 and 39 °C for 5 h, and the second step involved hydrolysis with a multi-enzyme pancreatin at pH 6.8 and 39 °C for 17 h.

Pahm [130] compared the use of an AD^II^ incubator with three Huang et al. [131] in vitro procedures using cellulase in the third step to that of Boisen and Fernandez [127] using Viscozyme or faecal inoculum in the third step. When using the AD^II^ incubator, these authors concluded that, of the three evaluated in vitro procedures, that of increasing the incubation length of the Boisen and Fernandez [127] using Viscozyme in the third step was the one that improved the sensitivity of the assay the most, and it provided a better R^2^ between the dry matter digestibility and apparent total tract digestibility of the gross energy, and between the dry matter digestibility and digestible energy, than the procedures that used cellulase or faecal inoculum.

Youssef and Kamphues [132] analysed a commercial swine diet, with lignocellulose A and B, by means an AD^II^ incubator, to determine its in vitro dry matter digestibility, using the fresh faeces of pigs as the inoculum source. The fermentation rates of the tested ingredients were evaluated using the caecum contents of swine as inoculum precursors, and these were then compared with that obtained with faeces inocula. The in vitro results were confirmed in vivo by testing the digestibility rate of the most digestible product of the lignocellulose ingredients. These authors found that the use of faeces/excreta liquor provided a valid estimate of the fermentation or digestibility of feeds, and they concluded that this procedure could be an effective way of approximating the digestibility of pig diets.

### 8.7. Dogs

Candellone et al. [71] recently performed in vitro analyses of dog pet food using the methods proposed by Hervera et al. [133] and Biagi et al. [134] utilizing Ankom bags and an AD^II^ incubator. They concluded that the two in vitro methods slightly overestimated the digestibility coefficients of the considered dog diets, when compared with the in vivo digestibility values. The in vitro method proposed by Hervera et al. [133] and utilized in this study yielded values closer to the in vivo results, in line with Hervera et al. [135], who showed a higher accuracy approach of in vivo crude protein apparent digestibility (R^2^ = 0.81) and in vivo digestible energy (R^2^ = 0.94), respectively.

## 9. Conclusions

This review summarised the use of the AD^II^ incubator in studies on digestibility in ruminants, as well as its extension to non-ruminants. From its introduction until today, the AD^II^ incubator has proved to be able to allow for the analysis of multiple feedstuffs, to improve the precision and reproducibility of an assay, and to reduce the time and costs of analysis. DMD values from AD^II^ and in situ techniques may be higher than those obtained in vivo [104], but both systems allow for the true digestibility of feedstuffs to be estimated, while the in vivo values only refer to the apparent digestibility.

Even though the use of the AD^II^ incubator is by now standardised, there is still a need for further research, as reported in Appendix A, to summarise some practical recommendations concerning the correct use of the AD^II^ incubator. To date, there are no standardised protocols for the collection, storage, and transportation of the rumen fluid or faeces. There is also a need to standardise the procedures for washing the bags after digestion. A major problem is the type of inoculum, which is the main source of variability of the system. Some performance metrics of the instrument (such as the reliability of the rotation mechanism of the jars) also require improvement.

The authors verified the need for caution when comparing data obtained from different methods, because they can yield different results [25]. Appendix A reports the variability of the AD^II^ instrument for 48 h of incubation, as well as the coefficient of variability (CV, %) within and between laboratory, runs, jars, and samples. Appendix A shows the correlation between AD^II^ and in vivo, in situ, and Tilley and Terry digestibility, as well as the respective linear equations.

The authors also verified that there is a lack of a standard terminology in studies and, as such, propose the use of the acronyms reported in Appendix A to make the language homogeneous.

Some potential developments and evolutions in the use of the AD^II^ incubator were also described. Created and developed for digestibility studies on ruminants, before being extended to monogastric and other non-ruminant species, this technology, in the future, could in fact be used for human digestibility studies or to obtain more detailed knowledge on the nutraceutical function of some feeds.

## Figures and Tables

**Figure 1 animals-10-00775-f001:**
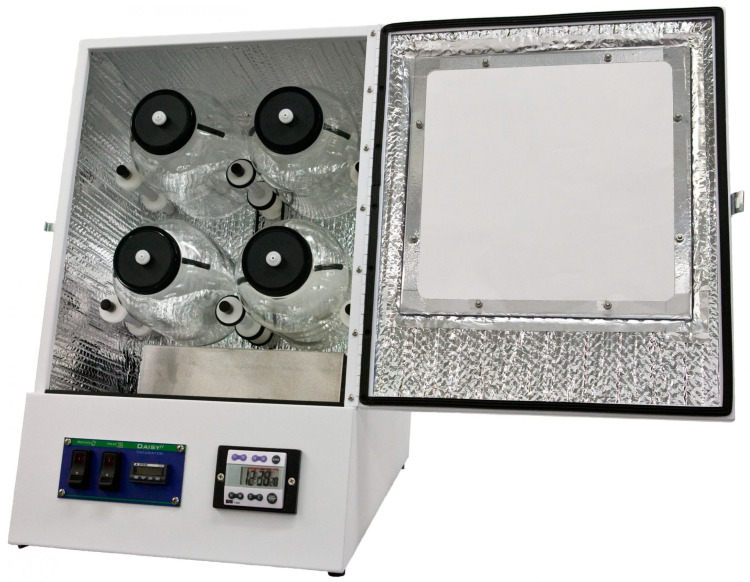
The Daisy^II^ incubator (Ankom Technology Corporation Fairport, New York, NY, USA).

**Table 1 animals-10-00775-t001:** Rumen fluid (RF) and fresh faeces (FF) inocula applied to the Ankom Daisy^II^ incubator (AD^II^).

Inoculum	Species	Sample Type	Notes	Ref.
RF	Dairy cattle	10 feeds	Variability of the dry matter digestibility for different donor cow diets as sources of inoculum	[22]
RF	Dairy cattle	By-products	AD^II^ vs. gas production, RF from slaughtered or cannulated cows	[41]
RF	Steers and Dairy cattle	Grains, total mixed ration, silages	Effect of the RF on the apparent and true dry matter digestibility DMD	[46]
RF	Sheep and Goats	Leaves, flowers and fruits of 5 browse plant species	Comparison of the true DMD and gas production kinetics with RF from animals fed the same diet	[47]
FF	Yaks	Forage produced at high altitude	Faeces vs. RF for a comparative digestibility trial	[57]
FF	Sheep and Camels	Fodder species from an arid environment	RF from sheep, and faeces from camels: comparative digestibility trial	[58]
FF-RF	Cattle	Feeds with different neutral detergent fibre (NDF) contents	NDF digestibility and undigested NDF measured with RF and 2 FF from cows fed different diets	[60]
FF-RF	Steers	35-day regrowth alfalfa hay	Comparative evaluation of the true dry matter digestibility; steers fed alfalfa or digit grass	[61]
FF	Horses	4 dietary treatments (hays or hay + oat)	Comparative evaluation of in vivo vs. in vitro DM and NDF digestibility	[64]
FF	Donkeys	7 common feeds for donkeys	Evaluation of the apparent and true DMD and neutral detergent fibre digestibility (NDFD) at 4 incubation times (30, 48, 60, and 72 h)	[65]

**Table 2 animals-10-00775-t002:** Sample size and bag type applied to the Ankom Daisy^II^ incubator (AD^II^).

Sample Size (g)	Bag Type	Sample Type	Notes	Ref.
0.25	F57	Forages and plant parts	Particle breakdown: 0.5, 1.0, and 1.5 mm	[23]
0.25	F57 and F58	Temperate and tropical grasses and legumes	uNDF after 240; effect of Na_2_SO_3_	[28]
0.25	Polyethylene polyester polymer bags	Low- and high-quality forages and grains	Different time delays and storage time between the collection of RF and the analysis	[30]
0.25	F57 and dacron bags (pore size: 0.30 and 0.50 μm)	Dried samples + 5 g glass balls	DMD of feed samples with alternatives to F57 and weighted to ensure submersion in the media	[79]
0.25	Dacron bags (pore size: 0.50 μm)	5 feeds + garlic or garlic oil vs. Monensin	Sheep RF, the effect of inclusion on organic matter digestibility (OMD)	[80]
0.30	F57	Triticale	Short and long (240 h) incubation times	[78]
0.50	5 × 3 cm pore size 0.45 μm	Pastures, forages and by-products	Comparison of in situ DM and NDF degradation kinetics	[33]
0.50	F0285 (pore size 0.25 μm)	Corn silage	Comparison of in vitro and in situ estimates of indigestible NDF at 2 fermentation end points (120 and 288 h)	[34]
0.25 0.50	F57	Triticale	Comparison of NDFD with 2 sample sizes	[76]
0.25 0.50	F57	7 feeds	Correlation with a conventional batch culture	[77]

**Table 3 animals-10-00775-t003:** Different buffer solutions used in in vitro digestibility trials with the Ankom Daisy^II^ incubator (AD^II^) in different animal species.

Buffer Solution References	AD^II^ References	Animal Species
[82]	[24]	Ruminants
	[61]	Ruminants
	[64]	Horses
	[65]	Donkeys
[83]	[72]	Ruminants
[84]	[85]	Ruminants
[86]	[25]	Ruminants

**Table 4 animals-10-00775-t004:** Precision and accuracy of the method using the Ankom Daisy^II^ incubator (AD^II^).

Parameters	Notes	Ref.
DMD and NDFD by means of the two-stage rumen fluid–pepsin technique (TT), the AD^II^ incubator and in situ; 0.25 and 0.50 sample size; 1 and 2 mm grinding size	The digestibility values estimated means of the by AD^II^ incubator and in situ techniques were correlated (R^2^ = 0.58–0.88) with values estimated by means of conventional in vitro and in vivo techniques. In most cases, the AD^II^ incubator and in situ techniques overestimated DMD and NDFD	[87]
In vitro DMD vs. Minson and McLeod technique [83]	Good reproducibility between and within the jars in the AD^II^ incubator	[72]
In situ (2 different filter bags) and TDMD (traditional bottles or the AD^II^ incubator)	The AD^II^ incubator underestimated the TDMD values but there was direct proportionality between the in situ and in vitro DMD values	[88]
NDFD of 18 hays	The variability was similar to that of some chemical analysis and lower than the in situ measurements	[89]
NDFD of 162 hays	Similar average values	[91]
NDFD and the associated calculated net energy lactation (NEl) of 10 fibrous feeds; 5 laboratories	Improved NDFD precision and improved accuracy and reproducibility of the calculated NEl for an extended fermentation time (48 h)	[92]
Validation of a modified TT by achieved by testing the repeatability and reproducibility of the new TT as well as the correlation with a previous version of the method	Good repeatability and reproducibility achieved when using the new version of the TT with the AD^II^ incubator; the same accuracy was achieved as that of the conventional method	[93]

**Table 5 animals-10-00775-t005:** Comparison of the Ankom Daisy^II^ incubator (AD^II^) with other digestibility methods.

Methods	Results (Referred to AD^II^ Technique)	Ref.
TT	True DMD: no differences	[97]
VS	True DMD considering 3 incubation times: the AD^II^ technique was less efficient but there were no significant differences	[98]
GVS	NDFD at different times: AD^II^ always lower than GVS; better results with F57 washed in acetone	[99]
GVS	NDFD: very few differences	[100]
TT, VS	Apparent and true DMD: significant differences	[102,103]
TSP	Intestinal digestibility of crude protein (R510 filter bags, up to 5 g sample): results closely results	[85]
TT	Validation of a modified TT with the AD^II^ technique	[93]
TT	Similar digestion values; the source of inoculum may affect DMD	[22]
TT	Apparent and true DMD, apparent and true OMD, NDFD. The TT gives more precise results but requires more labour	[23]
TT	Good agreement, but the AD^II^ technique gave higher values for some feeds	[73]
TT, gas production, in vivo	The results of 3 in vitro techniques (AD^II^, TT and gas production) were highly correlated with in vivo; AD^II^ technique is faster and more accurate	[105]
Minson and McLeod [83]	Higher digestibility values were obtained with AD^II^	[72]
In situ	NDFD was closely correlated	[92]
In situ	NDFD was 25–30% higher than in situ; a medium degree of correlation and low accuracy were achieved	[91]
In situ	Incubation at different times. The digestible NDF values were closely correlated at 48 h incubation, but the AD^II^ values of the NDFD were higher than the in situ values.	[89]
Different in situ and in vitro techniques	Lower reproducibility coefficients for AD^II^ than the other techniques; direct proportionality was observed between the in situ and in vitro DMD for different techniques	[88]
Batch culture, gas production	The AD^II^ dry matter digestibility values were higher than the gas production and batch culture values for longer incubation times than 12 h	[25]
In vivo, in situ, TT	The AD^II^ technique accurately predicted the in vivo DMD but overestimated in situ DMD; AD^II^ less accurately correlated with the TT	[87]

TT = Tilley and Terry; VS = Van Soest; GVS = Goering Van Soest, TSP = three-step procedure.

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
