# Peer review of "In Vitro Techniques Using the DaisyII Incubator for the Assessment of Digestibility: A Review"

_animals, 2020, doi:10.3390/ani10050775_

Round 1

Reviewer 1 Report

Dear Editor

The authors of the article "In vitro techniques using the DaisyII incubator for the assessment of digestibility: a review" accepted most of the suggestions and deeply revised the manuscript. The paper is clearly improved and enriched with explanatory tables.

Therefore, I believe that the article can be published in its current form after a few formal revision. My congratulations to the Authors for the work!

I list below some specific revisions of the manuscript:

Table 1s:

- Replace 2° with 2a

- The first equations at line 10b, the are missing the brackets: (W2*DM g/g)

- Define FM in the note.

- Clarify that w2 is the sample weight without the bag while w3 and w4 are the sample weight with the bag.

- Clarify how C1 is computed

Table 2s:

- What is the meaning of the variability factor "AD incubator"? is it the variability between runs?

Table 4s:

- In the column “Parameter” are missing the definitions of TDMD and dNDF Acronyms

Author Response

Dear Reviewer,

we have corrected the article according your suggestions.
All changes made to the manuscript have been written in red.

We would like to thank you for your careful analysis and your important suggestions that have improved this manuscript.

Yours sincerely,

Dr Sonia Tassone

Reviewer 2 Report

Comments to authors:

Thank you for the modifications made.  I hope you don't mind for a few more suggestions.

Line 19-21:  suggest to use “predictive capability”.  Suggestion: Modifications used by various researchers include the use of different inocula, different buffer solutions and different sample weights. Recently, attempts have been made to adapt the method to determine nutrient digestibility of feedstuff in non-ruminant animals including pets.

Line 41:  … ethically difficult in vivo method to “measure and “  predict “nutrient”digestibility in ruminants.

Line 40-44:  Suggest to delete the sentence “ Over 40 time, various techniques have been developed for this purpose” and go directly to line 41: The first method developed by Tilley and Terry (1) was a  two-stage rumen fluid-pepsin technique (TT). This method  provided satisfactory estimates of in vivo apparent digestibility [2], although some ….

Line 45:  This line can connect directly to line 44.  No need for a separate paragraph.

Line 47:  estimate the in vitro true digestibility of what? feedstuffs?? Grasses?  on the basis of the undigested cell-wall constituents.

Line 49-52: the paragraph is not consistent to the flow of the thought process and reads awkwardly. Suggest to delete this paragraph.

Line 53:  Suggestion: Other in vitro methods have been developed used to estimate the digestibility of feedstuff.  are based on the volume of gas  produced during a fermentation process. Menke and Steingass [15] proposed to measure  use of the gas produced during fermentation volume and feed composition data to estimate the energy content of feeds.

Line 85:  Suggest to delete the word “clear”

Line 86: delete “advantages”; redundant in the sentence

Line 87:  Terry and Van Soest methods. Because of the ADII design, it is capable of testing , and it is generally more suitable for a large number of samples [22-24].  \

Line 88:  Suggest to replace “cheap’ with “inexpensive”

Line 89-91: Just a suggestion “of several feedstuffs and diets [8,22]. ; However, compared to other techniques - such as the batch  culture technique (reference), the use of the  Ankom Gas Production System (reference) or the Rumen Simulation Technique (reference),  the ADII incubator was demonstrated to gives higher values at  different incubation times [25], but this also depends on the experimental design of the trial. “

Line 91: suggestion: One application of the ADII incubator concerns is  the estimation of a single point of neutral detergent fibre digestibility (NDFD

Line 93-94: suggestion: … digestibility (NDFD) at single time points a given in time endpoints (such as 30 or 48 h) using an Ankom Fiber Analyzer (Ankom Technology Corporation, Fairport, NY, USA) [26].  This line is already redundant as it already refers to ADII in line 91.

Line 94-96: Suggest to delete “The determined  accuracy and precision of NDFD after short- or long-term intervals is of considerable interest for research and industry.” Unless this refers to the “issues” mentioned in the next paragraph, in which case, this sentence can modified to fit in the next paragraph.

Line 97-105:  the sentence starts with “The attempts to address these issues have involved…”.  I am not clear on the “issues”.  Are the issues being referred to the accuracy and precision of conducting a long or short term NDFD?  Are the issues specific on the use of ADII or in-vitro methods in general.  Please clarify the “issues” in the beginning of the sentence so that the rest of the paragraph can connect to the issue stated.

Line 106: Suggest to move this sentence after line 93 as it fits well to the thought process.  Further suggest to modify as follows:

Recently, A recent application of the ADII incubator has been directed towards used for the in vitro long-term ruminal digestion (240 h) of undigested NDF (uNDF) [28]. To estimate the kinetics of NDF degradation, longer time intervals are essential specially when using complex models.  Complex models may require inputs of  fast, slow and indigestible NDF pools [35,36] which can be determined with ease when using ADII incubator.

Line 338-339: … “The different buffer solutions used for in vitro digestibility analysis are described in Table 3”.  Table 3 does not describe buffer solutions.  Column 1 in table 3 list down references that conducted studies on various buffer solutions. Column 2 in table 3 list down references using the ADII incubator. The other tables are well defined except for this table.  I think this tables need to be improved other than just listing references. 

Author Response

Dear Reviewer,

we have corrected the article according your suggestions.
All changes made to the manuscript have been written in red and the article has been reviewed by an English expert.

We would like to thank you for your careful analysis and your important suggestions that have improved this manuscript.

Yours sincerely,

Dr Sonia Tassone

Reviewer 3 Report

I have no further comments in this regard, since the authors have taken into account all my suggestions and have provided the appropriate additional clarifying information.

Author Response

Dear Reviewer,

We would like to thank you for your careful analysis and your important suggestions that have improved this manuscript.

Yours sincerely,

Dr Sonia Tassone

This manuscript is a resubmission of an earlier submission. The following is a list of the peer review reports and author responses from that submission.

Round 1

Reviewer 1 Report

Dear Editor

I revised the paper entitled: “In vitro techniques using the DaisyII incubator for the assessment of digestibility: a review”.

The manuscript reviews the literature of the last 25 years on the use of a commercial equipment designed to evaluate the rumen degradability and the gastrointestinal digestibility of feeds and diets. Moreover, the article lists the most recent applications of the instrument to different species and scientific objectives.

The topic and objectives of the work are of absolute scientific interest and provide a valuable contribution to the continuation of the research.

The article also describes the strengths and weaknesses of the method and why caution is needed when the digestibility values obtained with the different methodologies are compared.

Notwithstanding, in my opinion, the manuscript requires a deep revision at least for the following aspects:

1)        As noted by the Authors, the inoculum source “is important for in vitro fermentation studies, but it also represents the greatest source of uncontrolled variation in fermentation systems”. However, the characteristics and quality of the inoculum is not a specific problem of Daisy incubator but a general problem of all in vitro systems. Most of the literature reported in the manuscript does not refer to works that have used Daisy incubator (with only a few exceptions: ref. 49, 61, 31). This topic appears out of the aim of the manuscript and has been already addressed by other reviews (Mould et al. 2005. Yánez-Ruiz ˜ D.R 2016). For this reason, I suggest the Authors remove or strongly reduce this part. Alternatively, given the centrality of this topic for in vitro methods, and given that the discussion made by the Authors is interesting, I suggest the Authors revise the text indicating: A) the last reviews available on this topic B) the innovative aspects reported by the recent literature.

2)        The last part of the manuscript (chapter 8) appears as a simple list of the latest research. I’m aware that most of the research has extremely diversified objectives and it is not easy to obtain clear indications on the use of the Daisy, but the Authors should report the main results of the research and clarify the reasons why the instrument should be considered "valid".

3)        The Authors should report a list of practical recommendations on the use of Daisy (or a list of the main problems concerning the use of Daisy that require further study).

4)        Reduce at minimum the use of acronyms and uniform them over the manuscript to make the text more readable to non-experts.

Specific revisions:

L 84: In my knowledge, Ankom declares a pore size of F57 of 25 mm but the bag is composed of a material without a defined porosity. Please verify this central topic.

L96 Greater values of what? Moreover, the over or underestimation of digestibility values is a controversial topic as demonstrated by the discussion in chapter 7. Use caution in this statement

L214 The reference does not seem consistent with the text.

L255 The lower repeatability of NDFD compared to DMD and IVTD values has mathematical reasons and does not depend on the in vitro method or procedure.

L278 see L84

L317 Please check what Ankom declares for F58 porosity and if the information provides by Valentine et al. (2019) is reliable.

L366 Please report the bag size and porosity of R510 and the amount of feed sample per cm2 of bag.

L379 in vitro true analysis? or in vitro true digestibility values?

L384 As good practices, it can be suggested to put the different replicas of each food in different jars. Practical experience shows that the position of the jars respect to the heat source affects temperature, microbial activity, and degradability.

L388 Define "IS-nylon bag" and "CB" or avoid the use of acronyms.

L396 Revise the sentence “producibility (SR) and repeatability (Sr) of the between and within laboratory variability”

L407 see L384

L378, 398, 399, 404, 410 and L420. The use of different units of measurement confuses the reader. I recommend standardizing the information. Both SD and CV values can be useful because give a quantitative and proportional dimension of the variability. Also, consider the opportunity to organize the data in a table with other experimental information such as n. sample, etc.

L422 The literature report contradictory results on the relation between the different methods. The Authors should highlight this aspect and could synthesize the mean regressions and correlations values between AD vs other methods in a figure or a table reporting also the main methodological aspects that can affect the AD digestibility as: bag characteristics, the sample amount (g DM/cm2), rumen fluid source, particle size, apparent or true method, etc.

L432 AD and "In situ" methods overestimate DMD concerning what?

L439 It is necessary to specify “in DM”?

L438-450: the sentence is not easy to understand. Use always the acronyms (DMD and IVTD) or the terms apparent and true.

L453 “higher values ….” of what? DMD, IVTD,NDFD…

L486 what is TDMD? In this work was evaluated the true or apparent digestibility?

L499 e 504 In Spanghero et al (111) the “NDFD values” measured with the AD incubator were lower than those measured in situ but become higher (25-30%) when the in situ data were expressed as “effective NDFD”. Specify the difference between "NDFD at 48h" and "effective NDFD" or simplify the text removing the comparison with “effective NDFD” values.

L534, 539, 543,546, 550, 554, 556 The discussion should not only report simple statements of merit of the methodology (suitable, valid, adequate, comparable, similar, only comparable) but should report the experimental results that lead to these statements.

L552 Revise the sentence

L559 Reports the values of repeatability and reproducibility.

L569 Describe for which parameters the AD incubator has been “evaluated as a tool”

L581 “valid and accurate”, please report the values of accuracy and precision

L599 Specify why the differences between methods depend on the possibility of washing the crucibles?

L624 are the Ankom F56 bags?

L636-644. Give a dimension of the differences between methods

Conclusions: The overestimation AD methods respect in vivo data or others in vitro methods (TT or VS) cannot be generalized (see L655) but should be discussed and referred to the methodological procedure used as: bag characteristics, sample and bag size, rumen fluid, incubation time, particle size, feed characteristics, medium, …).

Moreover, the Authors should report some practical considerations on the use of daisy or at least a description of the main methodological aspects that must be reported in the M&M to allow a comparison between experiments.

Author Response

Dear Reviewer, 

we have revised the article according to your comments (Please see the attachment).

We take the opportunity to thank you for your careful analysis and your important suggestions.

We hope this modified version will fully satisfy you.

Yours sincerely,

Dr Sonia Tassone

Author Response

(The authors gave the same response as above.)

Reviewer 3 Report

Please allow me some suggestions that could be useful for the authors to improve the article

L21 “Monogastrics and pets” is not an adequate animal classification based in digestive traits. I suggest “non-ruminants, pets included”

L24 “new animal species” is an expression that could be misinterpreted. I suggest “other species different to ruminants”

L30 Some highlights or conclusions should be included in the abstract

L46 Please define “VS” the first time it was mentioned. Van Soest method I suppose

L51 Taking into account the aim of the RUSITEC system I would suggest “to maintain as far as possible the stability of the ruminal bacterial microbiota from the inoculum” rather than to talk about “normal microbial population”

L66 You have ignored the probably most important aspect of the RF Ankom system, such as automated and controlled gas release

L84 Please rewrite sentence to use F57 as a model and not as a known adjective: “Feed samples are weighted in filter bags (25μm pore size F57 Ankom bags, Ankom Technology Corporation Fairport, NY, USA)”

L85 Up to 92 bag per jar? It should be a mistake as Ankom recommends a maximum of 25

L93-96 In my opinion, it depends on the experimental design of the trial, since the main limitation of this technique in this sense lies in the common rumen culture medium for all treatments. This trait has been not sufficiently approached in this review

L106 The specific objective of the study of Raffrenato et al should be clarified

L116 Hypotheses instead of hypothesis

L141 “Digestive tract” instead of “gastro-intestinal”

L150-155 A very different approach with an extended point of view on this matter you can found in Ramos-Morales et al Anim Feed Sci Tech 2014 198, 57-66. Please consider it

L167 The conclusions of Lutakome et al are in total disagreement with Chaudry and the authors cited below. In my opinion the observation of those authors is not relevant in this review

L187-203 The conclusions of the recent research of Belanche et al (J Sci Food Agric 2019 99, 163-172) must be taken into account in this section

L193 Compared to?

L206-207 and L208-211 Despite the consideration of the high interest in their message, those paragraphs appear to be disjointed in the text

Section 3.2. In my opinion, the use of faeces as inoculum source must be briefly introduced to immediately describe and analyse only the articles that have been used this material in the DAII system.

L265 I cannot find any reference to the “sample preparation”. It should be better to enter the term “sample weight”

L355 Please clarify the meaning of the term "method". In my opinion, the type of buffer solution is not addressed now but the "methodology" to calculate digestibility

L365 To assist the reader, it is necessary to present the characteristics and standard use of Ankom R510 in-situ nylon bags rather than just a model reference into brackets

L478 Please rewrite the sentence

L481 Insert a line break after the dot

L495 Delete line break to merge this paragraph to the next

L519 In my opinion, it is not correct to differentiate monogastric and equine, so I suggest: In vitro digestibility using the DaisyII incubator for non-ruminants.

L544-546 Please specify if those experiment were conducted using DAII incubator

L564-570 Please indicate in which of the three studies an DAII incubator was used

L578 Were the faeces “directly” obtained from the camel rectal tract?

L584 With similar or better results than with faeces of small ruminants?

L636-639 Please include the conclusions of this comparative study

L642-643 The same as before: indicate some conclusions

L650 In general, the conclusions section should be rewritten since it does not summarize the review, it includes some aspects that were not previously discussed or included in the objective of the article (L657-659).

L652 “Non-ruminants” rather than “new species”

Author Response

(The authors gave the same response as above.)
